# High Resistance of SO$_2$ and H$_2$O over Monolithic Mn-Fe-Ce-Al-O Catalyst for Low Temperature NH$_3$-SCR

**Shijie Hao [1], Yandi Cai [1], Chuanzhi Sun [2], Jingfang Sun [1,\*], Changjin Tang [3,\*] and Lin Dong [1]**

[1] Jiangsu Key Laboratory of Vehicle Emissions Control, School of the Environment, Nanjing University, Nanjing 210093, China; shijie.hao@wfec.com.cn (S.H.); caiyd@smail.nju.edu.cn (Y.C.); donglin@nju.edu.cn (L.D.)

[2] College of Chemistry, Chemical Engineering and Materials Science, Shandong Normal University, Jinan 250014, China; suncz@sdnu.edu.cn

[3] School of Environment, Nanjing Normal University, Nanjing 210093, China

\* Correspondence: sunjf@nju.edu.cn (J.S.); tangcj@njnu.edu.cn (C.T.); Tel.: + 025-83594945 (J.S.); + 025-83594945 (C.T.)

**Abstract:** Monolithic Mn-Fe-Ce-Al-O catalyst with honeycomb cordierite ceramic as a carrier was reported for the first time for low temperature deNO$_x$ application. In the reaction of selective catalytic reduction (SCR) of NO with NH$_3$, a NO conversion of above 80% at 100 °C was obtained. Notably, the catalyst also showed excellent resistance against SO$_2$ and H$_2$O. About 60% NO conversion was maintained after successive operation in the mixed stream of SO$_2$ and H$_2$O for 168 h. The Brunner−Emmet−Teller (BET) measurement, SEM, EDS, thermogravimetric analysis (TG), FT-IR, and XPS results of the used catalysts indicated that certain amounts of ammonium sulfate was formed on the surface of the catalyst. XPS results revealed that partial of Fe$^{2+}$ was oxidized to Fe$^{3+}$ during the reaction process, and Fe$^{2+}$ species have strong redox ability, which can explain the decrease in activity after reaction. In addition, SO$_2$ and H$_2$O induced a transformation of Ce from Ce$^{4+}$ to Ce$^{3+}$ on the surface of the catalyst, which increased the amount of chemisorbed oxygen. Owing to these factors, the addition of Ce and Fe species contributes to excellent resistance of the catalyst to SO$_2$ and H$_2$O.

**Keywords:** honeycomb catalyst; Mn-Fe-Ce-Al-O; low temperature denitration; water and sulfur resistance

## 1. Introduction

As global air pollutants, nitrogen oxides (NO$_x$) emitted due to high-temperature combustion processes have caused diverse environment problems such as photochemical smog and ozone [1–3]. Generally, NO$_x$ emission contains two parts, i.e., stationary and mobile sources. Compared with mobile sources, the control of NO$_x$ emission from stationary sources is still a big challenge, particularly for non-electric fields including steel, cement, coking, glass, and other industries.

Selective catalytic reduction (SCR) in NO$_x$ with NH$_3$ is currently the main applied method for treatment of NO$_x$ from stationary sources. The compositions of commercial SCR catalysts are V$_2$O$_5$-WO$_3$/TiO$_2$ or V$_2$O$_5$-MoO$_3$/TiO$_2$, and the reaction temperature ranges from 290 to 380 °C [4], which adapts to the flue gas conditions of the power plant. However, for other industries like steel, etc., the temperature is usually lower than 250 °C, and the application of commercial SCR catalysts has been limited. On the other hand, the spent catalyst containing V$_2$O$_5$ is regarded as a hazardous waste, which makes the treatment of spent catalyst difficult. As such, the development of novel vanadium-free catalysts with high efficiency at low temperatures has become very urgent.

As a typical candidate for SCR, $MnO_x$-based catalysts have been widely investigated due to their superior low temperature activity. Tang et al. [5] prepared three unsupported $MnO_x$ catalysts with amorphous phase and found that the excellent catalytic activity was mainly due to their amorphous phase and high specific areas. Although $MnO_x$-based catalysts show excellent catalytic activity at low temperatures, their resistance to $SO_2$ and $H_2O$ deactivation still needs to improve. To address this point, adding new component is a widely used strategy. For example, Qi et al. [6,7] explored Mn-Ce and Mn-Fe/$TiO_2$ catalysts and found that the addition of iron oxide not only increased the NO conversion but also the resistance ability to $H_2O$ and $SO_2$. Dong et al. [8–11] synthesized a Mn-Ce/$Al_2O_3$ catalyst showing excellent $NH_3$-SCR performance at 100 °C. However, the catalyst in the above research was in powder state, which was much different to the monolithic catalyst used for practical production.

In our previous study, a recipe of Mn-Fe-Ce-Al-O was found to obtain good NO conversion efficiency. In this research, the monolithic catalyst was prepared by impregnation method with cordierite honeycomb carrier to verify their application potential. All the experiments were conducted on the denitration platform with a gas flow rate of 10 $m^3$/h. The NO conversion and $H_2O$/$SO_2$ resistance were investigated at 100 °C. According to the requirement of industrial catalyst testing, the continuous performance measurement lasted 168 h. The catalyst used for 168 h was thoroughly analyzed by a series of characterizations such as SEM, EDS, BET to know more details about the monolithic catalyst.

## 2. Results and Discussion

### 2.1. DeNO$_x$ Activity of Mn-Fe-Ce-Al-O Catalyst at Low Temperatures

In order to evaluate the catalytic performance of the monolithic Mn-Fe-Ce-Al-O catalyst at low temperatures, the testing temperature is confined to 80–130 °C. Figure 2a shows the activity test results. As well, the photos of reaction setup and honeycomb cordierite catalyst are provided in Figure 2b,c, respectively. It is found the NO conversion initially increases with reaction temperature and maintains above 80% in the temperature range of 100–130 °C. Particularly, more than 70% NO conversion can be achieved at the low temperature of 80 °C. The performance of monolithic catalyst is comparable to $MnO_x$-based catalyst in powder state. Hence, it is obvious that the monolithic Mn-Fe-Ce-Al-O catalyst exhibits excellent deNO$_x$ performance in the low-temperature region.

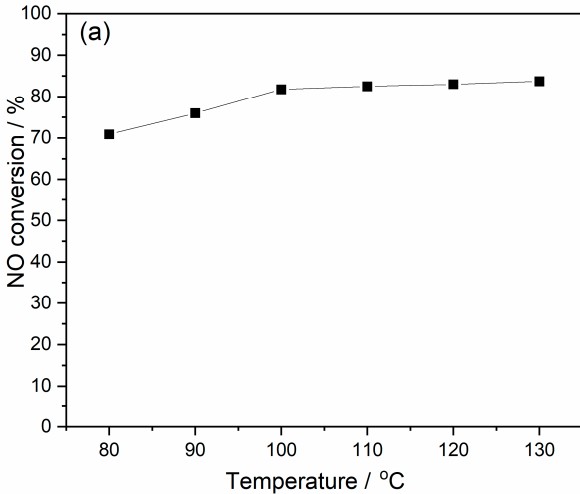

**Figure 1.** *Cont.*

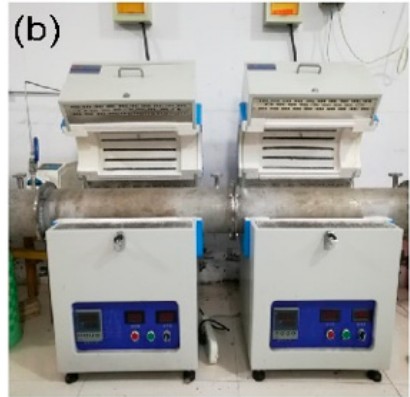 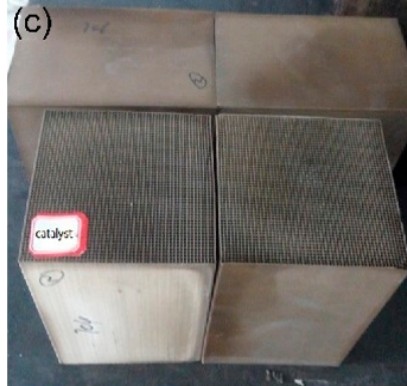

**Figure 2.** (**a**) The catalytic activity of monolithic Mn-Fe-Ce-Al-O catalyst as a function of reaction temperature. The photos of (**b**) reaction setup and (**c**) monolithic Mn-Fe-Ce-Al-O catalyst. Reaction conditions: 200 ppm NO, 200 ppm $NH_3$, air balance, gas hour space velocity (GHSV) = 1667 $h^{-1}$.

*2.2. Resistance of Mn-Fe-Ce-Al-O Catalyst to $SO_2$ and $H_2O$*

2.2.1. Resistance of Catalyst to $SO_2$

The reaction temperature was set at 100 °C and $SO_2$ concentration was controlled (50, 100, 200 ppm) to investigate the effect of $SO_2$ on the catalytic performance of Mn-Fe-Ce-Al-O catalyst. As shown in Figure 3, the NO conversion of Mn-Fe-Ce-Al-O catalyst decreases slightly after introduction of $SO_2$ with low concentration (50 and 100 ppm) and is maintained above 75% at 100 °C. The NO conversion of catalyst decreases obviously and finally stabilizes at 70% in the presence of 200 ppm $SO_2$. In comparison with the power $MnO_x$-based catalyst, it is found the NO conversion is similar, but the $SO_2$ resistance is much improved. The result suggests there may be some difference in the anti-$SO_2$ performance between power and monolithic catalyst. After removal of $SO_2$, the NO conversion is almost recovered to its initial level, indicating strong resistance ability to $SO_2$ of Mn-Fe-Ce-Al-O catalyst.

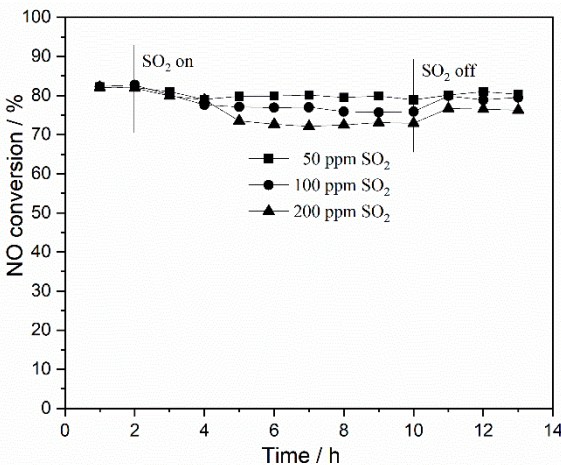

**Figure 3.** Effect of different concentration of $SO_2$ on the catalytic performance over Mn-Fe-Ce-Al-O catalyst at 100 °C. Reaction conditions: 200 ppm NO, 200 ppm $NH_3$, air balance, gas hour space velocity (GHSV) = 1667 $h^{-1}$.

2.2.2. Resistance of Catalyst to $H_2O$

The effect of different concentrations (4, 7, 10, 13 vol%) of $H_2O$ on NO conversion over Mn-Fe-Ce-Al-O catalyst were monitored at 100 °C, and the result was shown in Figure 4. The NO conversion of Mn-Fe-Ce-Al-O catalyst decreases after the addition of $H_2O$. With the increase in $H_2O$ (from 4 to 13 vol%), the denitration efficiency decreases gradually. When 4 vol% $H_2O$ was introduced

into the reactor, the NO conversion decreases from 80 to 70%, and it only obtains 45% when 13 vol% $H_2O$ is introduced. After $H_2O$ was shut off, NO conversion could be recovered. The results indicate that $H_2O$ deactivation is reversible, which could be caused by the competitive adsorption between $H_2O$ and $NH_3/NO$ [12].

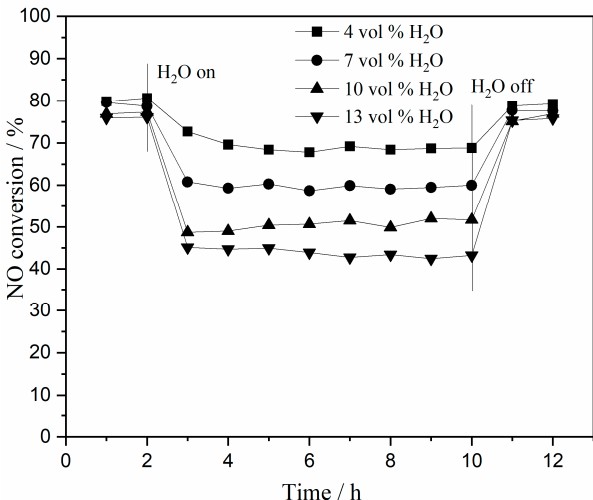

**Figure 4.** Effect of $H_2O$ concentration on deNO$_x$ performance over Mn-Fe-Ce-Al-O catalyst at 100 °C. Reaction conditions: 200 ppm NO, 200 ppm $NH_3$, air balance, gas hour space velocity (GHSV) = 1667 $h^{-1}$.

### 2.2.3. Resistance of the Catalyst against $SO_2$ and $H_2O$ for 168 h

An investigation on the stability of the Mn-Fe-Ce-Al-O catalyst in the presence of both $H_2O$ and $SO_2$ is presented in Figure 5. Under the employed reaction conditions (100 °C, without $H_2O$ and $SO_2$), 83% NO conversion is obtained. However, the NO conversion suffers from a small decline to 78% after adding 100 ppm $SO_2$ only and from 78 to 65% when 4 vol% $H_2O$ is also fed. It shows a decline trend gradually in the first 40 h. Then, the NO conversion maintains at about 60% for the next 120 h. After $H_2O$ shuts off, the NO conversion increases to 70%. Furthermore, the denitration efficiency is not changed after $SO_2$ removal. The results indicate that the inhibition effect is more obvious when the $SO_2$ and $H_2O$ coexists in the $NH_3$-SCR reaction. The NO conversion of the catalyst cannot recover to its initial level in the presence of $SO_2$ and $H_2O$ due to irreversible deactivation. It is reported that $SO_2$ can react with $NH_3$ to form sulfate species during the reaction in the atmosphere of $O_2$ and $H_2O$ [13]. To explore the deactivation mechanism of $H_2O$ and $SO_2$ on Mn-Fe-Ce-Al-O monolithic catalyst, the characterization of fresh and spent catalyst after 168 h reaction was carried out. For simplicity, the fresh catalyst is denoted as Z-Mn-Fe-Ce-Al-O, while the catalyst after 168 h reaction is marked as C-Mn-Fe-Ce-Al-O.

### 2.3. Physiochemical Characterization of Monolithic Mn-Fe-Ce-Al-O Catalyst before and after Reaction

### 2.3.1. XRD, BET, SEM, and EDS

Figure 6 shows the result of XRD patterns for catalyst before and after 168 h resistance test. The diffraction peaks at 28.95, 33.29, 47.49, 57.16, and 76.71° characteristic of $CeO_2$ (PDF# 34-0394) were obvious, suggesting that active substance mainly existed in the form of cubic fluorite type $CeO_2$ [9]. The typical peaks attributed to $MnO_x$ did not appear, indicating the crystallinity of $MnO_x$ was very low or in amorphous forms. There was no significant change in XRD patterns before and after the catalytic reaction, revealing that no crystal sulfate phase was formed or the particle size of surface sulfate was beyond the detection limit of XRD.

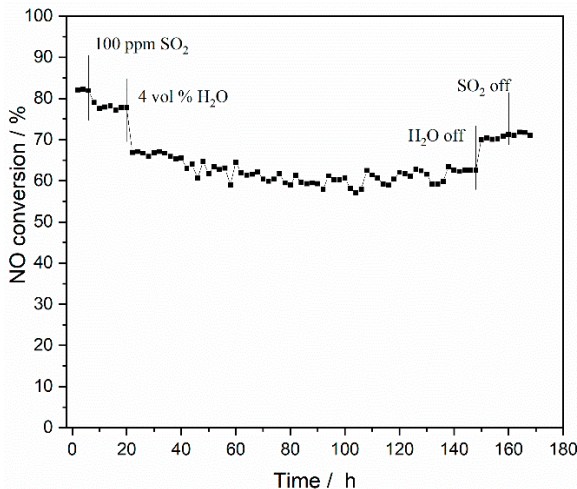

**Figure 5.** The long term resistance of Mn-Fe-Ce-Al-O catalyst against $SO_2$ and $H_2O$ at 100 °C. Reaction conditions: 200 ppm NO, 200 ppm $NH_3$, air balance, gas hour space velocity (GHSV) = 1667 $h^{-1}$.

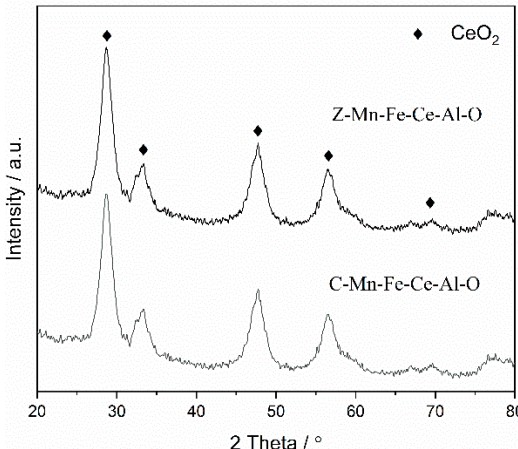

**Figure 6.** The XRD patterns of Mn-Fe-Ce-Al-O catalyst before and after 168 h resistance test.

The BET surface, pore volume, and pore size of Z-Mn-Fe-Ce-Al-O and C-Mn-Fe-Ce-Al-O catalysts are summarized in Table 1. The BET surface area and pore volume of the Z-Mn-Fe-Ce-Al-O catalyst are 77.90 $m^2$ $g^{-1}$ and 0.19 $cm^3$ $g^{-1}$, respectively. The values decreased to 52.34 $m^2$ $g^{-1}$ and 0.15 $cm^3$ $g^{-1}$ after the reaction. This indicates some surface species may be formed and cover on the surface of the catalyst, resulting in the pore blockage [14].

**Table 1.** The surface area and pore characterization of the catalysts before and after resistance test.

| Catalyst | $S_{BET}$ ($m^2$ $g^{-1}$) | Pore Volume ($cm^3$ $g^{-1}$) | Average Pore Diameter (nm) |
|---|---|---|---|
| Z-Mn-Fe-Ce-Al-O | 77.90 | 0.19 | 9.59 |
| C-Mn-Fe-Ce-Al-O | 52.34 | 0.15 | 11.21 |

Surface morphologies of Z-Mn-Fe-Ce-Al-O and C-Mn-Fe-Ce-Al-O catalyst are displayed in Figure 7. As can be seen, the surface of Z-Mn-Fe-Ce-Al-O catalyst is uniform. Comparatively, some aggregation and deposition can be observed on C-Mn-Fe-Ce-Al-O catalyst, which can reasonably explain the decrease in surface area of the catalyst. In addition, compositions of Z-Mn-Fe-Ce-Al-O and C-Mn-Fe-Ce-Al-O catalysts are further investigated by EDS analysis, and the results demonstrate that the main element of the particles on the surface of Z-Mn-Fe-Ce-Al-O catalyst is Mn, Ce, Fe, Al, O, C,

as shown in Table 2. However, the S signal is detected on C-Mn-Fe-Ce-Al-O catalyst, suggesting some sulfate species are probably deposited on catalyst surface.

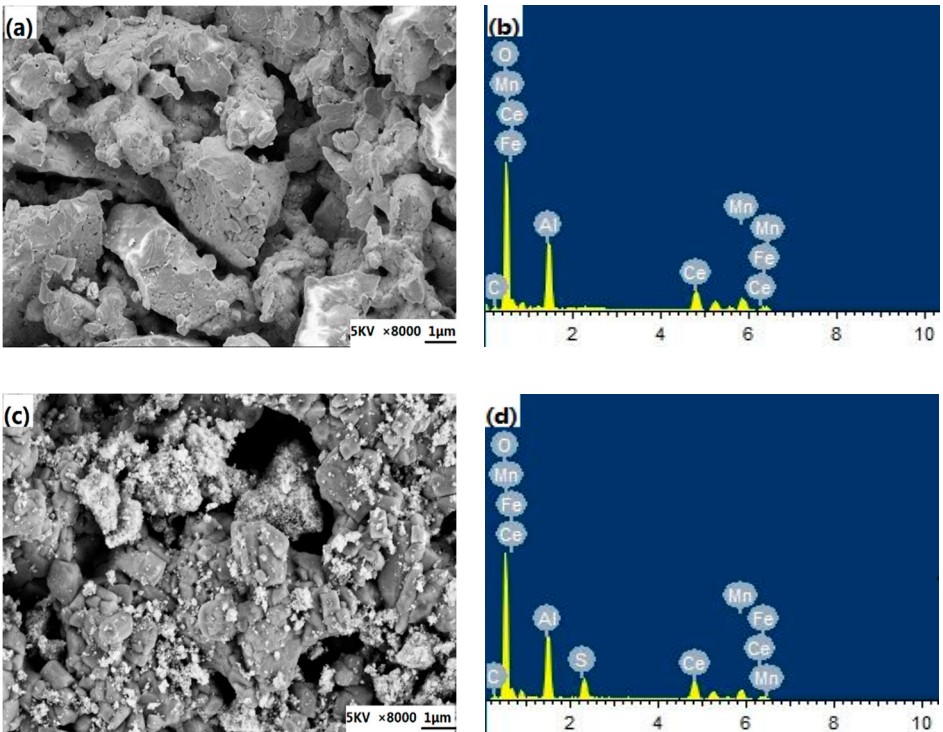

**Figure 7.** SEM photographs and EDS patterns of the catalysts before and after 168 h resistance test: (**a**) SEM of Z-Mn-Fe-Ce-Al-O, (**b**) EDS of Z-Mn-Fe-Ce-Al-O, (**c**) SEM of C-Mn-Fe-Ce-Al-O, (**d**) EDS of C-Mn-Fe-Ce-Al-O.

**Table 2.** Surface atomic content of the catalysts before and after 168 h resistance test (at%).

| Catalyst | O | C | Al | Mn | Ce | Fe | S |
|---|---|---|---|---|---|---|---|
| Z-Mn-Fe-Ce-Al-O | 64.55 | 9.11 | 12.44 | 5.98 | 6.18 | 1.73 | - |
| C-Mn-Fe-Ce-Al-O | 64.38 | 9.11 | 11.13 | 4.97 | 5.26 | 1.87 | 3.27 |

### 2.3.2. TG Analysis

Z-Mn-Fe-Ce-Al-O and C-Mn-Fe-Ce-Al-O catalysts were measured by TG to explore the thermal stability of sulfate species formed during the sulfation process. As shown in Figure 8, the weight losses of Z-Mn-Fe-Ce-Al-O and C-Mn-Fe-Ce-Al-O catalysts can be divided into three steps. Step 1 (25–200 °C) can be mainly assigned to the desorption of absorbed water on the catalysts [15], and C-Mn-Fe-Ce-Al-O catalyst shows much higher weight loss than Z-Mn-Fe-Ce-Al-O catalyst after being tested in presence of $H_2O$ and $SO_2$. Step 2 (200–400 °C) is mainly attributed to the decomposition of $(NH_4)_2SO_4$ (280 °C) and $NH_4HSO_4$ (390 °C) formed on the surface of catalysts [16]. The Z-Mn-Fe-Ce-Al-O catalyst shows no obvious decomposition or deposition behavior in the range of 200–400 °C. Furthermore, it can be seen that the C-Mn-Fe-Ce-Al-O catalyst displays further weight losses in step 3 (600–800 °C) and no similar phenomenon can be observed for the Z-Mn-Fe-Ce-Al-O catalyst. As reported elsewhere, the decomposition temperature of $Ce(SO_4)_2$ and $Ce_2(SO_4)_3$ was higher than 600 °C [17,18]. No obvious decomposition behavior of Z-Mn-Fe-Ce-Al-O and C-Mn-Fe-Ce-Al-O catalyst can be observed above 800 °C, indicating that no $MnSO_4$ (880 °C) species form in the catalyst [19].

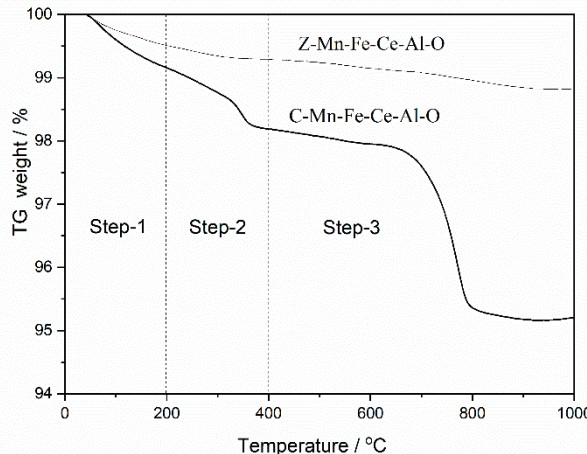

**Figure 8.** Thermogravimetric analysis (TG) curves of the Mn-Fe-Ce-Al-O catalysts before and 168 h resistance test.

### 2.3.3. XPS Analysis

Surface compositions of the fresh and spent catalysts are identified by XPS, and the results are shown in Figure 10 and Table 3. From Figure 10a,b, an obvious peak at 169 eV can be found for S 2p spectra of C-Mn-Fe-Ce-Al-O catalyst, which is identified as the characteristic peak of $S^{6+}$ state of $SO_4^{2-}$. No S 2p spectra can be observed for the Z-Mn-Fe-Ce-Al-O catalyst. It verifies that some sulfate species were formed on the surface of catalysts after $SO_2$ and $H_2O$ tolerance test [20].

Two main peaks due to Fe $2p_{1/2}$ and Fe $2p_{3/2}$ were shown in Figure 10c. The Fe 2p region consists of Fe $2p_{1/2}$ with a binding energy of about 725 eV and Fe $2p_{3/2}$ with a binding energy of about 711 eV. The Fe $2p_{3/2}$ peak was characteristic of a mixed valence iron system ($Fe^{3+}$ and $Fe^{2+}$). As shown in Figure 10c and Table 3, the ratio of $Fe^{2+}/Fe^{3+}$ of Z-Mn-Fe-Ce-Al-O catalyst (2.26) was higher than C-Mn-Fe-Ce-Al-O catalyst (0.56), which indicated that a part of $Fe^{2+}$ was oxidized to $Fe^{3+}$ during the reaction process [21–23].

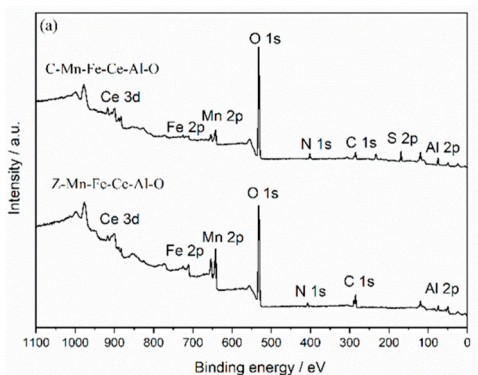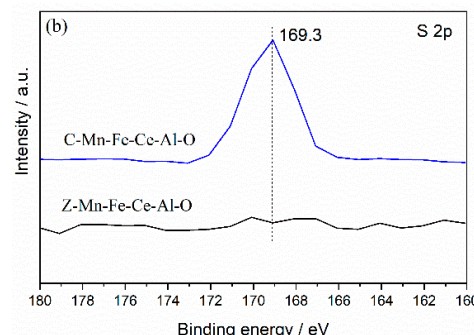

**Figure 9.** *Cont.*

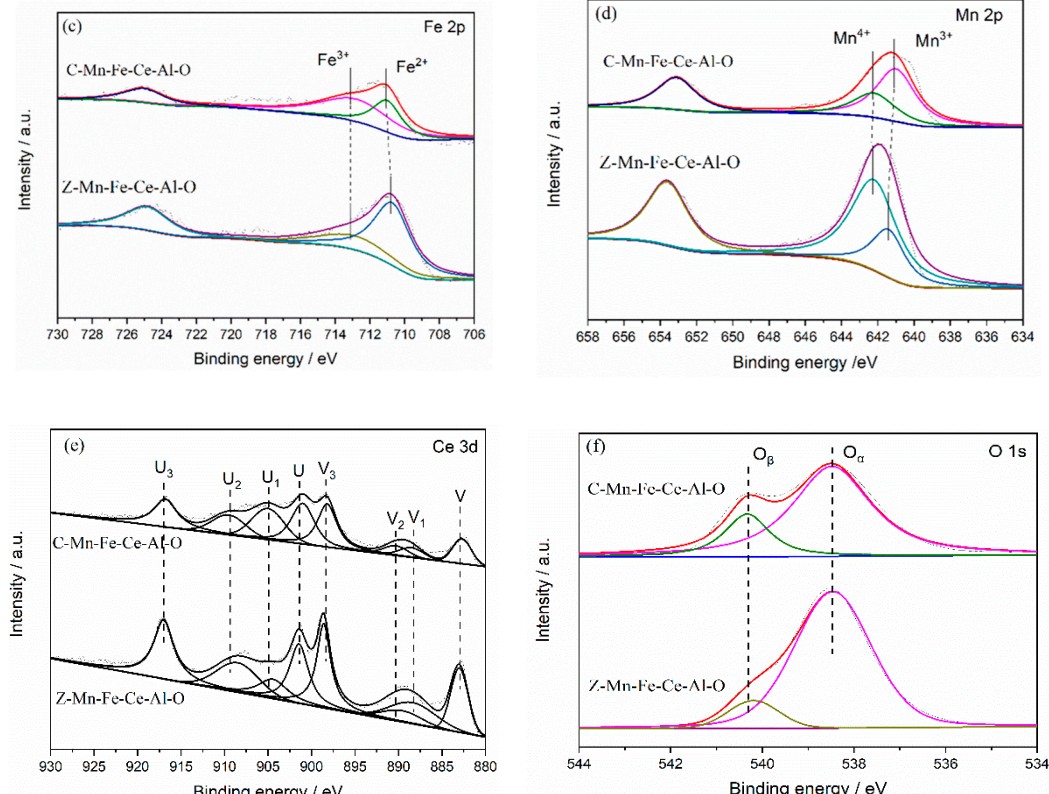

**Figure 10.** XPS spectra of the Mn-Fe-Ce-Al-O catalysts before and 168 h resistance test: (**a**) survey spectra, (**b**) S 2p, (**c**) Fe 2p, (**d**) Mn 2p, (**e**) Ce 3d, (**f**) O 1s.

**Table 3.** Surface atomic ratio of $Ce^{3+}/Ce^{4+}$, $Fe^{2+}/Fe^{3+}$, $O_\beta/(O_\alpha + O_\beta)$, $Mn^{4+}/Mn^{4+} + Mn^{3+}$ for Z-Mn-Fe-Ce-Al-O, and C-Mn- Fe-Ce-Al-O catalyst.

| Catalyst | $Ce^{3+}/Ce^{4+}$ | $Fe^{2+}/Fe^{3+}$ | $O_\beta/(O_\alpha + O_\beta)$ | $Mn^{4+}/(Mn^{4+} + Mn^{3+})$ |
|---|---|---|---|---|
| Z-Mn-Fe-Ce-Al-O | 0.22 | 2.26 | 0.19 | 0.72 |
| C-Mn-Fe-Ce-Al-O | 0.27 | 0.56 | 0.39 | 0.66 |

As shown in Figure 10d, the Mn 2p region consisted of spin-orbit doublet with Mn $2p_{1/2}$ having a peak at 651–655 eV and Mn $2p_{3/2}$ with a peak at 640–644 eV. The Mn $2p_{3/2}$ peak was fitted into two peaks referred to the $Mn^{4+}$ about 642 eV and the $Mn^{3+}$ about 641 eV [24]. To better understand the surface $MnO_x$ phases and their relative intensities, deconvolution of Mn 2p was conducted. Compared with Z-Mn-Fe-Ce-Al-O catalyst, the ratio of $Mn^{4+}/(Mn^{4+} + Mn^{3+})$ of C-Mn-Fe-Ce-Al-O catalyst decreased by 6%. Kapteijn et al. [25] reported that $Mn^{4+}$ had better redox ability than $Mn^{2+}$ and $Mn^{3+}$ of manganese oxide catalysts. Therefore, the NO conversion of catalyst decreased in the presence of $H_2O$ and $SO_2$.

The convoluted Ce 3d XPS spectra of Z-Mn-Fe-Ce-Al-O and C-Mn-Fe-Ce-Al-O catalyst are shown in Figure 10f. The Ce 3d peak was fitted into three peaks, which are denoted as U, $U_2$, $U_3$ and V, $V_2$, $V_3$, were attributed to $Ce^{4+}$ species, while $U_1$, $V_1$ were assigned to $Ce^{3+}$ species [26]. The intensities of $Ce^{3+}$ peaks increased after tested for 168 h, and the value of $Ce^{3+}/Ce^{4+}$ was increased from 0.22 to 0.27, which may suggest the occurrence of following reaction: $2CeO_2 + 3SO_2 + O_2 \rightarrow Ce_2(SO_4)_3$ [27]. The $SO_2$ may preferentially react with $CeO_2$, then the formation of the manganese sulfate can be greatly avoided. Similar phenomenon can also be found for other rare earth catalyst [28]. Therefore, the synergistic effect of Ce contributed to the excellent $SO_2$ and $H_2O$ resistance of the catalyst.

Figure 10g shows the O 1s XPS spectra of Z-Mn-Fe-Ce-Al-O and C-Mn-Fe-Ce-Al-O catalysts. The O 1s peaks could be fitted into two peaks referred to the chemisorbed oxygen at 538.5 eV (denoted as O$\alpha$) and the lattice oxygen at 540.3 eV (denoted as $O_\beta$). The ratio of $\mathbf{O_\beta}/(O_\alpha + O_\beta)$ over C-Mn-Fe-Ce-Al-O

catalyst (0.39) was higher than Z-Mn-Fe-Ce-Al-O catalyst (0.19), which was related to sulfate species formed during the SCR reaction for 168 h [29].

### 2.3.4. FT-IR Analysis

The FT-IR spectra of Z-Mn-Fe-Ce-Al-O and C-Mn-Fe-Ce-Al-O catalyst are shown in Figure 11. Both catalysts exhibit vibrations centered at 1638 and 3423 cm$^{-1}$. The band at 1638 cm$^{-1}$ features metal oxides [30], and the band at 3423 cm$^{-1}$ is due to the stretching vibration of surface O-H [31]. However, there are some new bands appearing at 1563, 1411, 1067, 1384, 1139 cm$^{-1}$ for the C-Mn-Fe-Ce-Al-O catalyst. The bands at 1563, 1411, and 1067 cm$^{-1}$ are from NH$_3$ adsorption on catalyst surface. The bands located at 1384 and 1139 cm$^{-1}$ could be ascribed to nitrate species and sulfate species [32], respectively. The results of FT-IR further confirm the appearance of sulfate, ammonium, and nitrate species on the surface of catalyst after 168 h test in the presence of H$_2$O and SO$_2$.

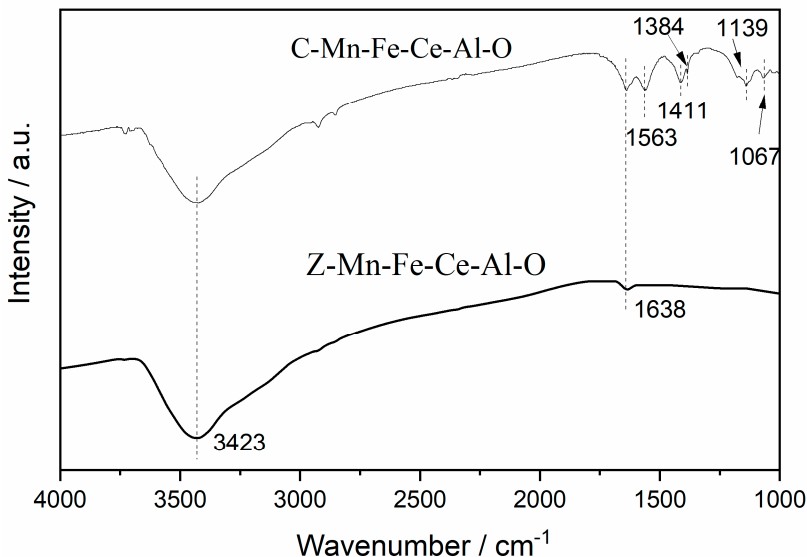

**Figure 11.** FT-IR spectra of the Mn-Fe-Ce-Al-O catalysts before and after 168 h resistance test.

## 3. Materials and Methods

### 3.1. Catalyst Preparation

The Mn-Fe-Ce-Al-O powder was prepared by co-precipitation method. Aqueous solutions of Al(NO$_3$)$_3$·9H$_2$O (Aladdin, Shanghai, China), Ce(NO$_3$)$_3$·6H$_2$O and Mn(NO$_3$)$_2$·4H$_2$O (Aladdin, Shanghai, China) were pre-mixed, and Fe(NO$_3$)$_3$·9H$_2$O (Aladdin, Shanghai, China) was added to the mixed solution. Then, ammonia solution (30 wt.%) was continuously added to the mixed solution with stirring until pH = 10. The precipitates were collected and washed with distilled water until pH = 7. The resulting mixture was dried at 110 °C for 12 h and calcined at 550 °C for 5 h. Finally, the catalysts were crushed and sieved to 150 mesh.

The monolithic Mn-Fe-Ce-Al-O catalyst was prepared by impregnation method with cordierite honeycomb as a carrier (100 mm × 100 mm × 150 mm). Firstly, the cordierite honeycomb was pre-treated by soaking in hydrogen peroxide (15 wt.%) for 6 h and dried at 120 °C for 10 h, and then calcined at 400 °C for 5 h. The honeycomb support has a channel density of 160 cells per square inch and a wall thickness of about 0.5 mm. Secondly, Polyvinyl alcohol (GRACE, Tianjin, China), Emulsifier (OP-10) (Aladdin, Shanghai, China) and sodium silicate (Aladdin, Shanghai, China) were dissolved in distilled water to form a solution and stirred for 1 h. Then, the prepared catalyst powder was slowly added into the solution with stirring for 30 min and wash-coated on the honeycomb with 10 wt.% loading. Finally,

the pretreated carrier was soaked in the mixed solution for 10 min. After impregnation, the monolithic catalyst was dried at 110 °C for 10 h and air calcined at 450 °C for 2 h.

### 3.2. Catalytic Activity Test

The denitration activity test was carried out on a self-designed reactor, consisting mainly of a gas distribution system, a catalytic reaction system, and a gas analysis system, as shown in Figure 12. The premixed simulated gases (200 ppm NO, 200 ppm $NH_3$, $SO_2$ (when used) and $H_2O$ (when used)) with the balance air were introduced into the reactor with a gas flow rate of 10 $m^3$/h, which heated by an air heater, and the gas space velocity was controlled at 1667 $h^{-1}$. The concentration of NO was measured using a flue gas analyzer (Testo-350, Lenzkirch, Germany).

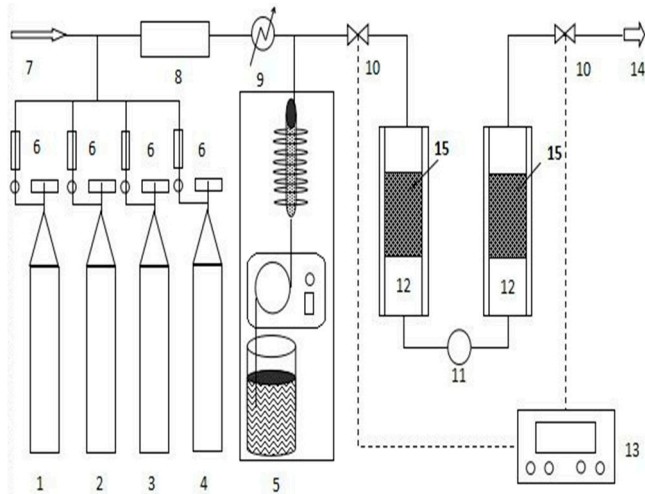

**Figure 12.** Schematic illustration of the activity evaluation system. 1—NO, 2—$SO_2$, 3—$N_2$, 4—$NH_3$, 5—vapor generator, 6—mass flowmeter, 7—air, 8—static mixer, 9—air heater, 10—sampling port, 11—temperature controller, 12—silica wool, 13—gas analyzer, 14—tail gas, 15—catalyst.

### 3.3. Catalyst Characterization

The X-ray diffraction (XRD) experiments were operated to determine the crystal phase structure (Rigaku5D/MAX RA, Tokyo, Japan). Cu K$\alpha$ radiation was employed, and the X-ray tube was operated at 40kV and 100 mA. The XRD spectra were recorded in the range of 10° < 2θ < 80° at 0.02° intervals. The BET-specific surface area of catalysts was determined by $N_2$ adsorption at 77 K using an automated gas sorption system (Micromeritics ASAP 2020 instrument, Shanghai, China). Prior to each analysis, the sample was degassed under vacuum at 300 °C for 4 h. Surface microstructure of the catalysts was analyzed by scanning electron microscopy (SEM, HITACHI SU8010 Tokyo, Japan). The X-ray photoelectron spectroscopy (XPS) experiments were carried out using ESCALAB 250Xi Thermol scientific X-ray photoelectron spectrometer system (Waltham, MA, USA) equipped with a monochromatic Al K$\alpha$ (hv = 1486.6 eV). An X-ray energy dispersive spectrometer (EDS, INCAOXFORD, Oberkochen, Germany) was used to analyze the surface element contents of catalysts. The energy resolution of the spectrum was 130 eV and the detection range of the elements was B~U. The thermogravimetric analysis (TG) was conducted using the thermal analyzer (SHIMADZU TA-60WS, New Castle County, DE, USA) to increase the temperature to 1000 °C under the inert gas condition. Fourier transform infrared (FT-IR) analysis was performed on a crushed scrap from the monolithic catalyst by using a Fourier transform infrared spectrometer (Nicolet iz10, in10, Thermol scientific, Waltham, MA, USA).

## 4. Conclusions

Monolithic Mn-Fe-Ce-Al-O catalyst was explored for the first time as an efficient catalyst for deNO$_x$ at low temperature. The NO conversion of the catalyst obtains more than 80% at the reaction temperature of 100 °C without SO$_2$ and H$_2$O. It also exhibits excellent resistance against SO$_2$, and the NO conversion can maintain at about 70% in the presence of 200 ppm SO$_2$ at 100 °C. The decreased activity caused by H$_2$O is reversible, and higher H$_2$O content induced a more obvious inhibition effect on the catalyst. It is worth noting that the catalyst can still obtain a NO conversion of 60% after 168 h resistance test against SO$_2$ and H$_2$O, indicating the excellent SO$_2$ and H$_2$O resistance of monolithic Mn-Fe-Ce-Al-O catalyst.

The characterization of BET, SEM, EDS, TG, FT-IR, and XPS indicated that ammonium sulfate was formed on the surface of catalyst after a 168 h resistance test against SO$_2$ and H$_2$O, which can explain the decrease in activity after reaction. In addition, the ratio of Mn$^{4+}$/(Mn$^{4+}$ + Mn$^{3+}$) of Mn-Fe-Ce-Al-O catalyst decreased after reaction, which is also one of the reasons for the decrease in activity. SO$_2$ and H$_2$O may induce a transformation of Ce from Ce$^{4+}$ to Ce$^{3+}$ on catalyst, indicating that SO$_2$ may preferentially react with CeO$_2$. Consequently, the formation of the manganese sulfate can be greatly avoided.

**Author Contributions:** J.S. and C.T. directed the research projects and supervised the work; S.H. conducted the experiments and summarized the data; C.S. and L.D. performed the refinement; Y.C. polished the English and provided the useful discussion. All authors discussed the results and contributed to the paper. All authors have read and agreed to the published version of the manuscript.

**Funding:** This work was financially supported by the National Natural Science Foundation of China (21707066, 21773106, 21976081, 21806077) and the National High Technology Research and Development Program of China (2015AA03A401).

**Conflicts of Interest:** The authors declare that they have no known competing financial interests or personal relationships that could have appeared to influence the work reported in this paper.

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
