# Peer review of "High Resistance of SO2 and H2O over Monolithic Mn-Fe-Ce-Al-O Catalyst for Low Temperature NH3-SCR"

_catalysts, doi:10.3390/catal10111329_

Round 1

Reviewer 1 Report

The manuscript presents the research about the novel catalyst of selective catalytic reduction of nitrogen oxides. The results of the work are interesting and undoubtedly give an added values to the findings published so far in the area of SCR catalysts.

Nevertheless, the manuscript is full of errors (especially language) that have to be corrected before publishing:

  1. First of all, language of the manuscript has to be improved. I would suggest to correct multiple language and grammar errors, for example in the following lines: [20] there should be no “the” before SO2 and H2O; [28] “emitted due to” rather than “resulted from”; [30] change stationary source and mobile source into “stationary and mobile sources”; [31] “mobile source” into “sources”; avoid colloquial expressions, such as “has long way to go” in lines [31-32]. Description of the literature review in the “Introduction” part has to be re-written in more readable way. Additionally, I would recommend to avoid citations that were published before 2010.
  2. In the “Materials and Methods” part: change “ammonia were” in line [67] into “ammonia solution was”; in line [72] change “*” into “x”; in line [74] “The honeycomb support (…)” – the sentence does not contain a verb; in line [78-79] change into “the monolith was wash-coated with 10 wt.% (…)”; the monolithic catalyst was dried, but what about calcination?
  3. In the “Catalytic activity test” part: in line [85] unnecessary bracket after “balance air”; in line [85-86] statement after comma without a verb; avoid mixed tenses, for example in line [85] and [86] – was and is in the same sentence; change the description of Figure 1.
  4. In the “Catalyst characterization” part: experiments cannot be “determined”, rather “carried out”, the specific physicochemical properties can be “determined”; change “observed” in line [98] into “analyzed”; change “on” in line [100] into “using”; line [102] one or more catalysts?; [106] the analysis was performed, not spectroscopy.
  5. In the “Results and discussion” part: paste the pictures below the results of NO conversion tests; in the description of Figure 2. the letters should be after the sentence; at the end of the description dot should not be indexed, the same in Figure 3., Figure 4. etc.; write more about the outcomes of NO conversion and stability in the presence of SO2 and H2O than only the observations; correlate the results with the properties of the catalyst; the descriptions of the experiments are not fully understandable and should be corrected; avoid naming figures as “Figure (…)” and than use “Fig. (…)” in text, such as in line [166]; improve the quality of Figure 6.; the units in all figures should be unified and taken in “/” or “()”; line [176] “Table . 1.” without the first dot; line [178] what kind of species?; Table 1. – change the font in the first line into one instead of two; line [188] change the sentence “there is … appears”; line [202] absorbed or rather adsorbed water?; avoid the word “obvious” in every sentence, especially in the description of TG; unify the font of the axes descriptions of XPS spectra; line [234] the spectra are shown instead of “were” shown, the same in line [254]; line [238] the occurrence instead of “occur”; line [255] – absorption peaks instead of “vibration absorptions”; line [256] stretching BOND; change “Figure 10.” Description from italics into straight letters; line [264] before and AFTER resistance test.
  6. In the “Conclusions” part: line [266] PRESENTED instead of present work; in line [267] the conversion cannot achieve.
  7. There is a considerable amount of citations that are not up to date. Try to find and cite literature that was published after 2010.

In summary, even though the research is interesting and the prepared catalyst could be promising, the authors must improve the whole manuscript to make it readable and well-understood by the readers. Otherwise, in the present form it is not suitable for publication.

Reviewer 2 Report

#1: The amount of S species in C-Mn-Fe-Ce-Al-O

The discussion of TG analysis (Section 3.3.2) is too qualitative. How much is the estimated S contents by decomposition of NH4HSO4 and Ce(SO4)2 or Ce2(SO4)3?

Do you have any other characterization data regarding total (including bulk) S content instead of EDX? For example, XRF and ICP is important characterization methods to discuss the bulk S content “quantitatively”. Please add the information about the amount of S adsorption after the aging treatment, which is quite important discussion for this article.

Reviewer 3 Report

In this paper, the authors report the activity of the Mn-Fe-Ce-Al-O in monolithic form to remove NOx of stationary sources at low temperatures via SCR. The topic is of high interest and there are several studies in the same line, although with the materials in powder form, so, the novelty is found in the configuration. I would suggest the publication of the manuscript after taking care of the following items:

  • Revise lines 74 and 75 since the sentence has not verb and is disconnected. “The honeycomb support with a channel density of 160 cells per square inch 75 and a wall thickness of about 0.5 mm”
  • In Catalyst characterization section, could you detailed the procedure to analyse the sample with FTIR? Is it was in powder or monolithic form? Is the sample pressed in disks or diluted?
  • In Figure 2.c, the insertion of the scale would be recommendable.
  • Sections 3.1 and 3.2 are too brief and descriptive, they contain the description of the results but there is no enough discussion. Please, increase the number of references and compare with other formulations in powder or monolithic form. Are the conversion values similar to those previously reported? Do H2O and SO2 affect in a similar way? Highlight more the benefits of this system.
  • In section 3.3.1, authors indicate that “This may be caused by some species formed on the surface of the catalyst, which blocked the pores of the catalyst”. I suggest authors to delve into this point and specify different alternatives.
  • In Table 2, Al, Mn and Ce contents decrease after reaction with the increase in sulfur, whereas Fe % remains constant. Is there an explanation to this effect?
  • Check the English throughout the manuscript, because there are grammar errors. Check the conjugation of subject and verb and the correct expression of the passive form; for example: nitrogen oxides “have” been important reactants in line 29, is “adapted” in line 36 or the application “has” been limited in line 38.

Round 2

Reviewer 1 Report

The manuscript in present for may be published.